# Proteomic Analyses of the G Protein-Coupled Estrogen Receptor GPER1 Reveal Constitutive Links to Endoplasmic Reticulum, Glycosylation, Trafficking, and Calcium Signaling

**DOI:** 10.3390/cells12212571

**Published:** 2023-11-03

**Authors:** Maryam Ahmadian Elmi, Nasrin Motamed, Didier Picard

**Affiliations:** 1Department of Cellular and Molecular Biology, School of Biology, College of Science, University of Tehran, Tehran 14155-6455, Iran; 2Département de Biologie Moléculaire et Cellulaire, Université de Genève, Sciences III, Quai Ernest-Ansermet 30, CH-1211 Genève, Switzerland

**Keywords:** GPR30, GPCR, APEX2-mediated proximity labeling, proteomics, interactome, maturation and trafficking, CLPTM1, PRKCSH, GANAB, STIM1

## Abstract

The G protein-coupled estrogen receptor 1 (GPER1) has been proposed to mediate rapid responses to the steroid hormone estrogen. However, despite a strong interest in its potential role in cancer, whether it is indeed activated by estrogen and how this works remain controversial. To provide new tools to address these questions, we set out to determine the interactome of exogenously expressed GPER1. The combination of two orthogonal methods, namely APEX2-mediated proximity labeling and immunoprecipitation followed by mass spectrometry, gave us high-confidence results for 73 novel potential GPER1 interactors. We found that this GPER1 interactome is not affected by estrogen, a result that mirrors the constitutive activity of GPER1 in a functional assay with a Rac1 sensor. We specifically validated several hits highlighted by a gene ontology analysis. We demonstrate that CLPTM1 interacts with GPER1 and that PRKCSH and GANAB, the regulatory and catalytic subunits of α-glucosidase II, respectively, associate with CLPTM1 and potentially indirectly with GPER1. An imbalance in CLPTM1 levels induces nuclear association of GPER1, as does the overexpression of PRKCSH. Moreover, we show that the Ca2+ sensor STIM1 interacts with GPER1 and that upon STIM1 overexpression and depletion of Ca2+ stores, GPER1 becomes more nuclear. Thus, these new GPER1 interactors establish interesting connections with membrane protein maturation, trafficking, and calcium signaling.

## 1. Introduction

The G protein-coupled estrogen receptor 1 (GPER1), also known as GPR30, was discovered in 1997. It belongs to the type A class of the rhodopsin-like subfamily of G protein-coupled receptors (GPCRs) [1]. Later, GPER1 was deorphanized by introducing estrogen as a ligand, based on two different assays and work from two independent research groups. First, SKBR-3 breast cancer cells, which endogenously express GPER1, and HEK293T cells transfected to express GPER1 exogenously were shown to bind to radioactive estrogen [2]. Second, it was shown that fluorescently labeled estrogen derivatives bind to and colocalize with GPER1 based on immunostaining and imaging by confocal microscopy. The latter study also revealed that GPER1 is predominantly localized in the membrane of the endoplasmic reticulum [3]. There is evidence that estrogenic signaling by GPER1 involves signaling to the epidermal growth factor receptor, activation of the mitogen-activated protein kinases (MAPKs) ERK1/2 [4], intracellular calcium mobilization [3,5,6,7], synthesis of phosphatidylinositol 3,4,5-trisphosphate in the nucleus [3], and activation of adenylyl cyclase through heterotrimeric G proteins [2,8]. Thus, GPER1 appears to mediate rapid nongenomic responses to estrogen [2,3,9]. In addition to the physiological estrogen 17β-estradiol (E2), it has been demonstrated that a variety of synthetic estrogenic chemicals can bind to and affect GPER1. Moreover, the selective agonist G-1 and the selective antagonists G-15 and G-36 have been developed for GPER1 [7,10]. It has been revealed that the assembly machinery of clathrin-coated pits can be recruited to GPER1 and lead to its sequestration and internalization in a β-arrestin-independent manner [6,11]. Moreover, GPER1 has attracted considerable interest because of its potential association with a variety of cancer types, including breast cancer [9,12,13,14,15,16,17].

Despite this substantial progress, some puzzling controversies about the mode of activation of GPER1, its function, and even its classification as a novel membrane-bound estrogen receptor persist. Attempts to confirm E2 binding to membranes from endothelial cells, which express GPER1, of mice knocked out for the two nuclear estrogen receptors (ERα and ERβ) failed. Moreover, E2 did not stimulate cAMP accumulation and phosphorylation of ERK1/2 in endothelial cells of ERα/ERβ double knockout mice [18]. E2 did not induce cAMP production or calcium release in ER-negative GPER1-expressing MDA-MB-231 cells and COS-7 cells transfected to express GPER1, even though the localization of GPER1 in the membrane of the endoplasmic reticulum of transfected COS-7 cells could be confirmed [19].

Doubts remain as to whether E2 binds to GPER1 directly. The ERα isoform ER-α36 was demonstrated to mediate nongenomic responses to estrogen through high-affinity binding to E2 and even the GPER1-specific agonist G-1 in SKBR-3 breast cancer cells and transfected HEK293T cells [20]. Others also reported that E2 and G-1 did not significantly elevate the phosphorylation levels of ERK1/2 in human breast cancer cell lines and bovine aortic endothelial cells [21]. Investigating the GPCR-mediated β-arrestin recruitment using the PathHunter β-arrestin recruitment technology [22] to identify cognate ligands for orphan GPCRs did not reveal recruitment of β-arrestin upon E2 stimulation of HEK293T cells transfected to express exogenous GPER1 [23]. To examine the in vivo effects of estrogenic responses mediated by GPER1, GPER1-deficient mice were generated. Their development of reproductive organs and functions was found to be normal. In addition, radioactive E2 failed to bind transfected cells expressing exogenous GPER1. The authors concluded that “the perception of GPR30 (based on homology related to peptide receptors) as an estrogen receptor might be premature and has to be reconsidered” [19]. The Leeb-Lundberg group recently reported that neither E2 nor G-1 could modulate the activity of GPER1 [24]. In contrast with previous reports, this group provided strong evidence for ligand-independent activity of GPER1 with a MAPK activity assay and using an assay exploiting the conformational changes of a Rac1 sensor. Moreover, in a multiplexed screen for novel ligands of GPCRs, HEK293T cells transiently or stably expressing GPER1 showed no responses to E2, the anti-estrogen 4-hydroxytamoxifen, or G-1 [25].

GPER1 has been demonstrated to be N-glycosylated in the N-terminal domain, which is predicted to be extracellular, and to contain a PDZ motif in the cytosolic C-terminal domain [26]. Several proteins have been identified to interact with GPER1 through its PDZ domain, such as SAP97, the SAP97-anchored protein AKAP5, PSD-95, PMCA4b, NHERF1, and RAMP3 [27,28,29]. It was shown that MAGUK and AKAP5 along with GPER1 assemble into a plasma membrane complex through the PDZ domain, but without G_i/o_. This GPER1 complex exhibited constitutive activity, which inhibited cAMP synthesis and maintained the receptor in the plasma membrane. The plasma membrane Ca^2+^-ATPase (PMCA) plays a key role in the depletion of cytoplasmic Ca^2+^ and in Ca^2+^ homeostasis. Intriguingly, it has also been reported that the complex formation with the PMCA subunit PMCA4b causes constitutive activation of GPER1, although in this case, E2 and G-1 treatment could further enhance GPER1 activity. In any case, these results suggested a possible crosstalk between calcium signaling and GPER1 activation [30] and illustrated the usefulness of identifying GPER1 interactors to promote our understanding of both ligand-dependent and constitutive activities of GPER1.

It is clear that novel tools or approaches are needed to clarify whether GPER1 directly binds to estrogen or any other ligands, and how signal transduction occurs. To clarify whether and how GPER1 may be activated by ligands, we set out to determine the GPER1 interactome more comprehensively using a combination of APEX2-mediated proximity labeling [31,32,33], immunoprecipitation (IP), and mass spectrometry (MS). Knowing the GPER1 interactome might help understand its functions, resolve its signal transduction pathways, and facilitate the development of effective therapies, for example, in the context of cancer [34,35,36]. Here, we provide an initial investigation of the possible roles of PRKCSH, CLPTM1, GANAB, and STIM1, which were among the top hits of our GPER1 proteomics. Our results further emphasize that the prevailing concept of GPER1 activation and function may need to be reconsidered and subjected to additional experimental scrutiny. Ultimately, our results might help to deorphanize or redeorphanize GPER1.

## 2. Materials and Methods

### 2.1. Cell Culture

Human embryonic kidney 293T cells (HEK293T; ATCC reference CRL-3216) and HeLa cells (ATCC reference CCL-2), which are established GPER1-negative cell lines, were cultured in Dulbecco’s Modified Eagle’s Medium (DMEM) supplemented with 10% fetal bovine serum (FBS) and 1% penicillin/streptomycin. For transfection experiments, cells were maintained in phenol red-free DMEM supplemented with 10% FBS, 2 mM L-glutamine, and 1% penicillin/streptomycin for 24 h. Then, cells were plated in the same type of medium and transfected with expression vectors for proteins as indicated, and with the empty expression vector pcDNA3.1(+) as negative control. For this, DNA was mixed with PEI MAX (1:4, in µg) (Polysciences Inc. # 24765-100; from Chemie Brunschwig, Basel, Switzerland); added to the cells in phenol red-free DMEM supplemented with 10% FBS, 2 mM L-glutamine, and 1% penicillin/streptomycin; and left overnight. The next day, the medium was discarded, and fresh DMEM supplemented with 5% charcoal-treated FBS, 2 mM L-glutamine, and 1% penicillin/streptomycin (hormone-deprived medium) was added. Forty-five hours after transfection, cells were starved in serum-free DMEM supplemented with 2 mM L-glutamine and 1% penicillin/streptomycin for 3 h, before they were treated with ligands as indicated.

### 2.2. Plasmids and Cloning

To generate a plasmid for expression of the GPER1-APEX2 fusion protein, the linker sequence (GGATCCGGTGGAAGTTCTGGCGGTTCAAGT), which codes for the polypeptide sequence GGSSGGSS, was included in the forward primer used for PCR amplification of the APEX2 coding sequence from plasmid pcDNA3 APEX2-NES (here referred to as F-APEX2-NES; a gift from Alice Ting; obtained from Addgene as plasmid #49386) [32]. The coding sequence for 3xFLAG-GPER1 was amplified from plasmid 3xFLAG-GPER1 [37] (here referred to as F-GPER1) and combined with the above-mentioned APEX2 sequence to generate plasmid F-GPER1-APEX2 for the expression of N-terminally 3xFLAG-tagged GPER1 fused to APEX2. To generate an expression vector for GPER1 with an N-terminal HA tag (HA-GPER1), the sequences for the HA tag were added within the forward primer to amplify the coding sequence of GPER1 from plasmid 3xFLAG-GPER1. The following constructs were also used: pcDNA3 APEX2-NLS (here referred to as V5-APEX2-NLS; a gift from Alice Ting; obtained from Addgene as plasmid #124617) [38]; mCherry-STIM1, mCherry-STIM1 (1-241), and mCherry-STIM1 (1-154) (gifts from Nicolas Demaurex, Université de Genève) [39]; Clptm1-hSyn-CFP and U6-scramble-hSyn-CFP (gifts from Ann Marie Craig, University of British Columbia) [40]; FLAG-PRKCSH (here referred to as F-PRKCSH; a gift from Dr. Gu-Choul Shin, The Catholic University of Korea) [41]; and a Rac1Cluc sensor plasmid (a gift from Dr. Björn Olde, Lund University, Sweden) [24]. See Appendix A for a schematic representation of plasmids used in this study.

### 2.3. Immunofluorescence Staining and Microscopy

For imaging experiments, plasmids for expression of F-GPER1, F-GPER1-APEX2, F-APEX2-NES, V5-APEX2-NLS, mCherry-STIM1, Myc-CLPTM1, and F-PRKCSH were transfected into HeLa cells. HeLa cells on coverslips were fixed with 4% paraformaldehyde dissolved in phosphate-buffered saline (PBS) at room temperature for 15 min. Then, cells were washed 3× with cold PBS and permeabilized in blocking buffer (0.1% Triton X-100, 3% bovine serum albumin (BSA) in 1× PBS) at room temperature for 30 min. Cells were washed again 3× with cold PBS. To detect F-GPER1, F-GPER1-APEX2, F-APEX2-NES, and F-PRKCSH, cells were incubated with a mouse anti-FLAG antibody (Invitrogen, cat. No. R960-25, 1:1000 dilution; from Thermo Fisher Scientific, Plan-les-Ouates, Switzerland), for V5-APEX2-NLS with a mouse anti-V5 antibody (GeneTex, cat. No. GTX1179, 1:1000 dilution; from LubioScience, Zürich, Switzerland), for mCherry-STIM1 with a rabbit antiserum against mCherry (Life Technologies, Thermo Fisher Scientific, Plan-les-Ouates, Switzerland, cat. No. PA534974, 1:1000 dilution), and for CLPTM1 with a rabbit antiserum against CLPTM1 (Abcam, Lucerna-Chem AG, Luzern, Switzerland, cat. No. ab174839, 1:1000 dilution) for 2 h at room temperature. After washing 4× with cold PBS for 5 min each, cells were incubated with secondary Alexa Fluor 488-conjugated goat anti-mouse IgG (Invitrogen, cat. No. A-11001, 1:2000 dilution; from Thermo Fisher Scientific, Plan-les-Ouates, Switzerland) or goat anti-rabbit IgG (H + L) (Highly Cross-Adsorbed Secondary Antibody, Alexa Fluor Plus 555, from Thermo Fisher Scientific, Plan-les-Ouates, Switzerland, cat. No. A32732, 1:2000 dilution) for 1 h at room temperature. Cells were then washed 4× with cold PBS for 5 min each. After incubation with 4′,6-diamidino-2-phenylindole (DAPI) diluted in PBS (1:50,000 dilution) for nuclear staining, cells were washed 4× for 5 min each with cold PBS. Coverslips were mounted on slides with Fluoromount-G (SouthernBiotech, from BioConcept, Allschwil, Switzerland) for imaging with a fluorescence microscope (Axiovert 100 from Zeiss, Feldbach, Switzerland).

### 2.4. Rac1 Sensor Assays

HEK293T cells were seeded in 6 cm plates in phenol red-free DMEM supplemented with 10% FBS and cotransfected with the Rac1Cluc sensor plasmid along with expression vectors for F-GPER1, F-GPER1-APEX2, or F-APEX2-NES overnight. Twenty-four hours after transfection, transfected cells were seeded into white-bottom 96-well plates (20,000 cells/well) and grown in a hormone-deprived medium. Cells were starved in DMEM without phenol red and serum for 3 h before treatment with ligands and then incubated with 60 µL/well DMEM containing 1% (*w*/*v*) D-luciferin (sodium salt; Cayman Chemical # 14682-500, from AdipoGen AG, Fuellinsdorf, Switzerland) for 2–3 h in the dark at 37 °C with 5% CO_2_ in a humidified incubator. Luminescence was then measured at different times with a Cytation 3 Imaging Reader (BioTek, from Bucher Biotec, Basel, Switzerland).

### 2.5. Proximity Labeling Experiments

HEK293T cells were plated in 15 cm plates and transfected with expression vectors for F-GPER1-APEX2 and F-APEX-NES with the PEI MAX reagent (1:4) (Polysciences Inc. #24765-100, from Chemie Brunschwig, Basel, Switzerland) overnight. The day after transfection, the medium was discarded, and the hormone-deprived medium was added. After 24 h, cells were starved in serum-free DMEM for 3 h. Then, cells were incubated with biotinyl tyramide (Chemodex, St. Gallen, Switzerland, # B0270-M100) at a final concentration of 500 µM for 45 min. After 35 min, 100 nM 17β-estradiol (E2) was added to one of the plates with F-GPER1-APEX2-expressing cells. At 44 min, H_2_O_2_ was added to 1 mM final concentration, and plates were agitated for 1 min at room temperature. Then, the medium was discarded, and the reaction was quenched three times with quencher solution (10 mM sodium azide, 10 mM sodium ascorbate, and 5 mM Trolox (AdipoGen AG, Fuellinsdorf, Switzerland, # AG-CR1-3639-G005)). The cells were harvested by scraping and centrifugation at 3000× *g* for 10 min at 4 °C.

### 2.6. Pull-Down of Biotinylated Proteins for MS

The cell pellets were lysed at 4 °C with RIPA lysis buffer (100 mM Tris-HCl pH 7.5, 150 mM NaCl, 1% Triton X-100, 0.5% sodium deoxycholate, 0.1% SDS) supplemented with 1× protease inhibitor cocktail (Thermo Fisher Scientific, Plan-les-Ouates, Switzerland, #78429), 1 mM PMSF, 10 mM sodium azide, 10 mM sodium ascorbate, and 5 mM Trolox. Lysates were clarified by centrifugation at 13,000× *g* for 10 min at 4 °C. Streptavidin magnetic beads (Pierce, Thermo Fisher Scientific, Plan-les-Ouates, Switzerland, catalog no. 88817) were washed twice with RIPA buffer, and 1.5 mg of each whole cell lysate (WCL) was incubated with 60 µL magnetic beads in 1.5 mL microcentrifuge tubes with rotation overnight at 4 °C. The beads were subsequently washed twice with 1 mL RIPA lysis buffer, once with 1 mL 1 M KCl (dissolved in diethylpyrocarbonate (DEPC)-treated water), once with 1 mL 0.1 M Na_2_CO_3_ (dissolved in DEPC-treated water), once with 1 mL 2 M urea in 10 mM Tris-HCl pH 8.0, and finally twice with Dulbecco’s PBS without calcium and magnesium (Gibco, Thermo Fisher Scientific, Plan-les-Ouates, Switzerland, cat. No. 14190-094). Streptavidin magnetic beads were transferred to fresh tubes after the last washing step and stored at −20 °C until 60% and 40% of these samples were used for MS and control immunoblots, respectively. The MS experiment and analysis was performed with 3 biologically independent replicates.

### 2.7. Immunoblot Analysis of Biotinylated Proteins

For these experiments, biotinylated proteins were eluted from streptavidin magnetic beads by boiling in 75 µL 3× protein loading buffer supplemented with 20 mM dithiothreitol (DTT) and 2 mM biotin. Thirty µg of each WCL and 20 µL of each eluate were separated on 4–12% SDS-PAGE gels (NuPAGE 4–12% Bis-Tris 1.0–1.5 mm Mini Protein Gels # NP0322BOX, from Thermo Fisher Scientific, Plan-les-Ouates, Switzerland). Gels were transferred to nitrocellulose membranes (GVS Life Science) and blocked with blot blocking buffer (3% (*w*/*v*) BSA and 0.1% Tween-20 in Tris-buffered saline) at 4 °C overnight. To reveal biotinylated proteins, the blot was incubated with streptavidin-HRP in blot blocking buffer (1:3000 dilution, Thermo Fisher Scientific, Plan-les-Ouates, Switzerland, cat. No. 21126) at room temperature for 1 h and washed with 0.1% Tween-20 in Tris-buffered saline (TBST) 5× for 5 min. For displaying the APEX2 proteins themselves from WCL, validation of candidates from WCL, and also pull-down elutions, the blots were blocked in 5% non-fat dry milk in TBST for 30 min at room temperature. For blots for F-GPER1, F-GPER1-APEX2, and F-APEX2-NES; a mouse anti-FLAG antibody (Invitrogen, Thermo Fisher Scientific, Plan-les-Ouates, Switzerland, cat. No. R960-25, 1:1,000 dilution), for IKBIP, a rabbit anti-IKBIP polyclonal antibody (Thermo Fisher Scientific, Plan-les-Ouates, Switzerland, cat. No. PA565219); and for LAMB1, a rabbit anti-LAMB1 polyclonal antibody (Thermo Fisher Scientific, Plan-les-Ouates, Switzerland, cat. No. PA527271) was incubated overnight at 4 °C. As appropriate for the primary antibody, blots were incubated with an HRP-conjugated goat anti-mouse secondary antibody (Thermo Fisher Scientific, Plan-les-Ouates, Switzerland, cat. No. 31430) or an HRP-conjugated goat anti-rabbit secondary antibody (Thermo Fisher Scientific, Plan-les-Ouates, Switzerland, cat. No. 31460), both at 1:10,000 dilutions, in TBST for 1 h at room temperature. After three washes with TBST, blots were developed using the WesternBright chemiluminescent substrate (Advansta #K-12045-D50, from Witec AG, Luzern, Switzerland) and imaged using an Amersham ImageQuant 800 biomolecular imager.

### 2.8. IP of F-GPER1 and F-GPER1-APEX2 for MS

HEK293T cells were cultured in DMEM supplemented with 10% FBS and 1% penicillin/streptomycin. Cells were plated on 15 cm cell culture plates and transfected with plasmids for expression of F-GPER1 or F-GPER1-APEX2 with the PEI MAX reagent (1:4) overnight. The day after transfection, the medium was discarded, and a hormone-deprived medium was added. After 24 h, cells were starved for 3 h in serum-free DMEM (to ensure the same conditions as for the proximity labeling experiment) and then harvested by scraping and lysed with lysis buffer (10 mM Tris-HCl pH 7.5, 50 mM NaCl, 1 mM EDTA, 10% glycerol, 10 mM Na-molybdate), supplemented with 1× protease inhibitor cocktail (Thermo Fisher Scientific, Plan-les-Ouates, Switzerland, #78429). Lysates were clarified by centrifugation at 13,000× *g* for 10 min at 4 °C and immunoprecipitated in parallel with an anti-FLAG antibody and a mouse IgG control antibody overnight at 4 °C. Protein G Dynabeads (Thermo Fisher Scientific #10009D) were washed twice with the same lysis buffer and added to lysates for 2 h at 4 °C. Then, magnetic beads were subsequently washed 5× with lysis buffer, each time with 1 mL for 10 min; transferred to fresh tubes; and stored at −20 °C until further processing for liquid chromatography–tandem MS (LC-MS/MS).

### 2.9. Immunoblot Analysis of Immunoprecipitated Samples

Magnetic beads were mixed with NuPAGE LDS Sample Buffer (Thermo Fisher Scientific, Plan-les-Ouates, Switzerland, # NP0008), incubated for 5 min at 95 °C in a heat block, and loaded onto a Tris-glycine 4–12% SDS-polyacrylamide gel (NuPAGE ™ 4–12% Bis-Tris 1.0–1.5 mm Mini Protein Gels # NP0322BOX, from Thermo Fisher Scientific, Plan-les-Ouates, Switzerland). The gel was transferred to a nitrocellulose membrane (GVS Life Science) and blocked in 5% non-fat dry milk in TBST for 30 min at room temperature. The nitrocellulose membranes were then incubated overnight at 4 °C, for F-GPER1, F-GPER1-APEX2, and F-PRKCSH, with a mouse anti-FLAG antibody (Invitrogen, Thermo Fisher Scientific, Plan-les-Ouates, Switzerland, cat. No. R960-25, 1:1000 dilution); for mCherry-STIM1, with a rabbit antiserum against mCherry (Life Technologies, Thermo Fisher Scientific, Plan-les-Ouates, Switzerland, cat. No. PA534974, 1:1000 dilution); for STIM1, with a rabbit antiserum against STIM1 (Life Technologies, cat. No. MA119451, 1:1000 dilution); for CLPTM1, with a rabbit antiserum against CLPTM1 (Abcam, Lucerna-Chem AG, Luzern, Switzerland, cat. No. ab174839, 1:1000 dilution); for Myc-CLPTM1, with an antibody against Myc (mouse monoclonal hybridoma supernatant, a gift from Thomas Kreis) at a 1:1000 dilution; and for GANAB, with a rabbit antiserum against GANAB (Life Technologies, cat. No. ab179805, 1:1000 dilution) for 2 h at room temperature. Further steps were as described above.

### 2.10. General Data Analyses

Data analysis was conducted using GraphPad Prism (version 9).

### 2.11. Proteomics

#### 2.11.1. Preparation of Peptides of Proximity Labeling Samples

For proximity labeling samples, beads were digested following a modified version of the iST method [42,43] (named miST method). Twenty-five µL of miST lysis buffer (1% sodium deoxycholate, 100 mM Tris pH 8.6, 10 mM DTT) was added to the beads. After mixing and dilution 1:1 (*v*:*v*) with H_2_O, samples were heated for 5 min at 95 °C. After digestion with 0.5 µg of trypsin/LysC mix (Promega AG, Dübendorf, Switzerland, #V5073) for 1 h at 25 °C, sample supernatants were transferred to new tubes. Beads were washed with 50 µL of miST buffer diluted 1/1 in H_2_O, and supernatants were pooled with the previous ones. Reduced disulfides were alkylated by adding 25 µL of 160 mM chloroacetamide (32 mM final) and incubating for 45 min at 25 °C in the dark. Samples were then digested overnight at 25 °C with 1.0 µg trypsin/LysC mix. To remove sodium deoxycholate, two sample volumes of isopropanol containing 1% trifluoroacetic acid (TFA) were added to the digests, and the samples were desalted on a strong cation exchange (SCX) plate (Oasis MCX; Waters Corp., Milford, MA, USA) by centrifugation. After washing with isopropanol/1% TFA, peptides were eluted in 200 µL of 80% acetonitrile, 19% water, 1% (*v*/*v*) ammonia; dried by centrifugal evaporation; and resuspended in 0.05% TFA, 2% (*v*/*v*) acetonitrile.

#### 2.11.2. LC-MS/MS Analyses of Proximity Labeling Samples

Data-dependent LC-MS/MS analyses of proximity labeling samples were carried out using a Fusion Tribrid Orbitrap mass spectrometer (Thermo Fisher Scientific, Plan-les-Ouates, Switzerland) interfaced through a nano-electrospray ion source to an Ultimate 3000 RSLCnano HPLC system (Dionex, from Thermo Fisher Scientific, Plan-les-Ouates, Switzerland), via a FAIMS interface (Thermo Fisher Scientific, Plan-les-Ouates, Switzerland). Peptides were separated on a reversed-phase custom-packed 45 cm C18 column (75 μm ID, 100 Å, Reprosil Pur 1.9 µm particles, Dr. Maisch, Germany, from Morvay, Basel, Switzerland) with a 4-90% acetonitrile gradient in 0.1% formic acid (total time 140 min). Full MS survey scans were performed at 120,000 resolution. A data-dependent acquisition method controlled by Xcalibur software version 4.3 (Thermo Fisher Scientific, Plan-les-Ouates, Switzerland) was set up that optimized the number of precursors selected (“top speed”) of charge 2^+^ to 5^+^ while maintaining a fixed scan cycle of 1.0 s per FAIMS compensation voltage (CV) (−40, −50, −60 V). Peptides were fragmented by higher-energy collision dissociation (HCD) with a normalized energy of 32%. The precursor isolation window was 1.6 Th, and the MS2 scans were performed in the ion trap. The *m*/*z* of fragmented precursors was then dynamically excluded from selection during 60 s. Data files were analyzed with MaxQuant 1.6.14.0 [44] incorporating the Andromeda search engine [45]. Cysteine carbamidomethylation was selected as a fixed modification, while methionine oxidation and protein N-terminal acetylation were specified as variable modifications. The sequence databases used for searching were the human reference proteome based on the UniProt database (www.uniprot.org, version of 6 June 2021, containing 79,057 sequences) and a “contaminant” database containing the most usual environmental contaminants and enzymes used for digestion (keratins, trypsin, etc.). Mass tolerance was 4.5 ppm on precursors (after recalibration) and 20 ppm on HCD fragments. Both peptide and protein identifications were filtered at 1% FDR relative to hits against a decoy database built by reversing protein sequences. All subsequent analyses were performed with the Perseus software package (version 1.6.15.0) [46]. Contaminant proteins were removed, and LFQ intensity values [47] were log2-transformed. After assignment to groups, only proteins quantified in at least 3 samples of one group were kept. After the imputation of missing values (based on normal distribution using Perseus default parameters), *t*-tests were carried out among all conditions, with permutation-based FDR correction for multiple testing (*q*-value threshold < 0.05). The difference of means obtained from the tests was used for 1D enrichment analysis on associated GO/KEGG annotations as described [48]. The enrichment analysis was also FDR-filtered (Benjamini–Hochberg, *q*-value < 0.02). Data were visualized in R.

#### 2.11.3. Preparation of Peptides of IP Samples

For immunoprecipitates, 40 µL of 2× loading buffer was added to the dried beads, and the samples were heated for 5 min at 95 °C. Twenty-seven µL was loaded onto a 12% SDS-polyacrylamide gel, run for about 3.0 cm, and stained with colloidal Coomassie blue. Gel lanes between 15 and 250 kDa were excised into 5 pieces and digested with sequencing-grade trypsin as described [49]. Extracted tryptic peptides were dried and resuspended in 0.05% TFA, 2% (*v*/*v*) acetonitrile.

#### 2.11.4. LC-MS/MS Analyses of IP Samples

The analyses were carried out on a “timsTOF Pro” mass spectrometer (Bruker, Bremen, Germany) interfaced through a nanospray ion source (“captive spray”) to an Ultimate 3000 RSLCnano HPLC system (Dionex, from Thermo Fisher Scientific, Plan-les-Ouates, Switzerland). Peptides were separated on a reversed-phase custom-packed 45 cm C18 column (75 μm ID, 100 Å, Reprosil Pur 1.9 µm particles, Dr. Maisch, Germany, from Morvay, Basel, Switzerland) at a flow rate of 0.250 µL/min with a 2–27% acetonitrile gradient in 93 min followed by a ramp to 45% in 15 min and to 90% in 5 min (total method time: 140 min, all solvents contained 0.1% formic acid). The data-dependent acquisition was carried out using a standard method with trapped ion mobility spectroscopy (TIMS) and parallel accumulation serial fragmentation (PASEF) [50] with ion accumulation for 100 ms for each survey MS1 scan and the TIMS-coupled MS2 scans. The duty cycle was kept at 100%. Up to 10 precursors were targeted per TIMS scan. Precursor isolation was performed with a 2 or 3 *m*/*z* window below or above *m*/*z* 800, respectively. The minimum threshold intensity for precursor selection was 2500. If the inclusion list allowed it, precursors were targeted more than one time to reach a minimum target total intensity of 20,000. The collision energy was ramped linearly based uniquely on the 1/k0 values from 20 (at 1/k0 = 0.6) to 59 eV (at 1/k0 = 1.6). The total duration of a scan cycle, including one survey and 10 MS2 TIMS scans, was 1.16 s. Precursors could be targeted again in subsequent cycles if their signal increased by a factor of 4.0 or more. After selection in one cycle, precursors were excluded from further selection for 60 s. Mass resolution in all MS measurements was approximately 35,000. See above for further details about data analysis. Data files were analyzed with MaxQuant 1.6.14.0 [44] incorporating the Andromeda search engine [45]. Cysteine was selected as a fixed modification, while methionine oxidation and protein N-terminal acetylation were specified as variable modifications. The sequence databases used for searching were the human reference proteome based on the UniProt database (www.uniprot.org, version of 3 September 2020, containing 75,796 sequences) and a “contaminant” database containing the most usual environmental contaminants and enzymes used for digestion (keratins, trypsin, etc.). Mass tolerance was 10 ppm on precursors (after recalibration) and 25 ppm on MS/MS fragments. Both peptide and protein identifications were filtered at 1% FDR relative to hits against a decoy database built by reversing protein sequences. All subsequent analyses were performed with the Perseus software package (version 1.6.15.0) [46]. Contaminant proteins were removed, and IBAQ [43] intensity values were log2-transformed. Missing values were imputed based on normal distribution using Perseus default parameters. The difference of means was used for 1D enrichment analysis on associated GO/KEGG annotations as described [48]. The enrichment analysis was also FDR-filtered (Benjamini–Hochberg, *q*-value < 0.02).

## 3. Results

### 3.1. Localization of F-GPER1 and Fusion Protein F-GPER1-APEX2

In view of using the proximity labeling system with APEX2 [31,32,33], an improved version of ascorbate peroxidase, its impact on GPER1 localization and function needed to be evaluated. To verify that the fusion of APEX2 to the C-terminus of GPER1 did not change its subcellular localization, we performed an immunofluorescence (IF) experiment with transfected HeLa cells. We exogenously expressed F-GPER1, the F-GPER1-APEX2 fusion protein, and, for comparison, FLAG-tagged APEX2 with a nuclear export signal (F-APEX2-NES) and V5-tagged APEX2 with a nuclear localization signal (V5-APEX2-NLS). Using an anti-FLAG antibody, IF images showed the same localization for F-GPER1 and F-GPER1-APEX2. Our images are compatible with the previously reported localization of GPER1 to the membrane of the endoplasmic reticulum [3]. As expected, F-APEX2-NES and V5-APEX2-NLS displayed diffuse cytoplasmic and nuclear staining, respectively (Figure 1).

### 3.2. Conformational Rac1 Sensor Assay Indicates Constitutive F-GPER1 and F-GPER1-APEX2 Activities

To confirm that the fusion of APEX2 to the C-terminus of FLAG-tagged GPER1 (F-GPER1-APEX2) did not change GPER1 activity, we used a split click-beetle luciferase-based Rac1 sensor as a proxy of GPER1-mediated activation of phosphoinositide 3-kinase and its downstream effector Rac1. The readout with the Rac1 sensor encoded by plasmid Rac1Cluc is based on a conformational change of the hybrid protein, which allows the intramolecular reconstitution of a functional luciferase [24,51] (Appendix A). HEK293T cells transiently expressing either F-GPER1 or F-GPER1-APEX2 displayed constitutively decreased luminescence compared to cells transfected with an empty expression vector as negative control. Surprisingly, the luminescence did not change significantly upon the addition of 100 nM E2 (Figure 2). Our results agree with the aforementioned report indicating constitutive activity for GPER1 [24] and suggest that the fusion of APEX2 to the C-terminus of GPER1 may not perturb its activity.

### 3.3. Identification of Potential GPER1 Interaction Partners by Proximity Labeling

The results of our activity assay indicated that GPER1 activity may not always be regulated by ligands. Identifying GPER1 interactors might set the basis for resolving some of the unknowns of GPER1 signaling and functions. We began our analysis by overexpressing proteins of interest in HEK293T cells and using the APEX2-mediated proximity labeling technique [31,32,33]. The APEX2 moiety of the F-GPER1-APEX2 fusion protein catalyzes the conversion of biotin phenol to biotin-phenoxyl (BP) radicals in the presence of hydrogen peroxide (H_2_O_2_). BP radicals can covalently conjugate biotin to endogenous proteins that are in proximity to F-GPER1-APEX2. Biotin-labeled proteins can then be purified by streptavidin-coupled magnetic beads for further analysis by gel electrophoresis and blotting, or by MS. We used F-APEX2-NES as negative control. Since F-GPER1-APEX2 and F-APEX2-NES might themselves become biotinylated, we probed pulled-down samples with an anti-FLAG antibody. F-GPER1-APEX2 could specifically be detected in the samples treated with BP + H_2_O_2_ (Figure 3A). The lower band of about 70 kDa corresponds to the expected full-length form of the fusion protein, whereas the much larger bands of ≥140 kDa could represent glycosylated forms and/or dimers or hetero-oligomers. Albeit clearly present in the input extract, biotinylated and pulled-down F-APEX2-NES could not readily be detected, possibly because F-APEX2-NES is itself a poor biotinylation substrate and/or because its 1xFLAG yielded a weaker overall signal (Figure 3A). Probing the pulled-down samples with streptavidin-HRP, we found that cells expressing F-GPER1-APEX2 yielded a distinct pattern of biotinylated proteins, which was not obviously altered by treating the cells with E2 for 10 min before triggering the proximity labeling. In contrast, this pattern was different from the one obtained with the negative control F-APEX2-NES (Figure 3A, right part). Two prominent bands (marked with an asterisk in Figure 3A, right part) are known to be endogenously biotinylated proteins, independently of APEX2 expression [52,53]. Paralleling these results and our aforementioned activity assay, the proteomic analysis showed that there were hundreds of cellular proteins that were differentially biotinylated upon expression of F-GPER1-APEX2 compared to F-APEX2-NES, irrespective of E2 treatment. Furthermore, we could not detect any biotinylated proteins indicative of a statistically significant difference between untreated and E2-treated cells expressing F-GPER1-APEX2 (see Appendix A). Moreover, the Venn diagram and the volcano plots (Figure 3B–D) are visual and quantitative depictions of these findings obtained by applying stringent cutoffs. A subset of proteins is specifically indicated in the volcano plots.

### 3.4. Wide Variety of Potential Interaction Partners for GPER1

We performed a gene ontology (GO) analysis to identify the enrichment of protein hits in the GO sets “Cellular Component”, “Molecular Function”, and “Biological Process” (Figure 4A–C). We separately compared F-GPER1-APEX2 without and with E2 to F-APEX2-NES as negative control. Consistent with our IF results (Figure 1), the GO analysis indicated that many of the significantly enriched hits are associated with the membrane and lumen of the endoplasmic reticulum, the endoplasmic reticulum–Golgi intermediate compartment, focal adhesion, and cell–substrate junction (Figure 4A). The GO analysis revealed a significant enrichment of numerous proteins linked to cadherin binding and focal adhesion in cells expressing F-GPER1-APEX2 without and with E2 (Figure 4A,B). The enrichment of proteins associated with GO terms mentioned above is visually presented as a hierarchically clustered heat map in Figure 5A. As examples, we validated the enrichment of the IKBKB-interacting protein (IKBIP), a biomarker related to the epithelial–mesenchymal transition [54], and of LAMB1, a crucial component of the extracellular matrix [55], by immunoblotting of proteins biotinylated upon expression of F-GPER1-APEX2 without and with E2 induction (Figure 5B). The highly significant and specific biotinylation of protein kinase C substrate 80K-H (PRKCSH) and neutral α-glucosidase AB (GANAB), which are the regulatory and catalytic subunits of α-glucosidase II, respectively, is illustrated as part of the heat map of Figure 5A. The following proteins associated with the GO terms “ERAD pathway”, “Unfolded protein response”, and “Glycosylation” were also significantly enriched in F-GPER1-APEX2-expressing cells with and without E2 compared to the negative control: Mannosyl-oligosaccharide glucosidase (MOGS), UDP-glucose:glycoprotein glucosyltransferase 1 (UGGT1), calnexin (CANX), calreticulin (CALR), ERP29, PRKCSH, and GANAB, which are involved in glycoprotein maturation and trafficking toward the cell surface, were the top hit candidates based on our statistical analysis (Figure 5A), which is noteworthy considering that GPER1 is a glycoprotein [26,56]. Surprisingly, we failed to see an enrichment of proteins related to the plasma membrane, such as G_α_ subunits of heterotrimeric G proteins, which are well-described interactors of GPCRs in their inactive mode [31]. Our results even show a depletion of GNAS (G_αs_) by about 3-fold by comparison with the negative control (Figure 3C,D and Figure 5A; Appendix A). Our findings also demonstrate significant enrichment of nucleoporin 210 (NUP210) of the nuclear pore complex, and torsins, which are membrane proteins of the endoplasmic reticulum and the nuclear envelope [57] (Appendix A).

### 3.5. Identification of Potential GPER1 Interactors by IP-LC-MS/MS

To complement our proximity labeling approach, we aimed to identify proteins that are part of a complex with GPER1 by IP. Therefore, we immunoprecipitated F-GPER1 and F-GPER1-APEX2, which were exogenously expressed in HEK293T cells, using an anti-FLAG antibody. A non-immune IgG was used in parallel as a control IP. Immunoblotting of immunoprecipitates and the input WCL with an anti-FLAG antibody confirmed the specific presence of F-GPER1 and F-GPER1-APEX2 in IP and input samples (Figure 6A). The subsequent proteomic analysis by LC-MS/MS revealed a profile of proteins that are specific and associated with both F-GPER1 and F-GPER1-APEX2 (Figure 6B; Appendix A). Intriguingly, the comparison of the proteins obtained by proximity labeling–LC-MS/MS and IP-LC-MS/MS, based on a fold change of ≥ 3.16 (log_10_ = 0.5) with *p*-values ≤ 0.05 and iBAQ scores ≥ 3.4, respectively, showed that about 10% of all hits are common to both (Figure 6B). Since we obtained these hits (73 proteins; see Appendix A) with two orthogonal methods, we consider them as high-confidence direct or indirect GPER1 interactors. Of these 73 proteins, many are related to the GO terms “maturation and trafficking”, “endoplasmic reticulum membrane”, “proteasome”, and “ERAD pathway” (Figure 6C). The keywords “membrane protein maturation and trafficking” and “endoplasmic reticulum membrane” caught our attention since intracellular routing and maturation of GPER1 are still poorly understood. We decided to focus on the following proteins for further validation: CLPTM1, PRKCSH, and GANAB in the context of GPER1 maturation and trafficking, and STIM1 as a calcium sensor and key player in calcium signaling [58,59,60,61] (Figure 6C).

### 3.6. F-GPER1 Interacts with Myc-CLPTM1 in HEK293T Cells

We performed a co-IP experiment with transfected HEK293T cells to verify the expected interactions between F-GPER1 or F-GPER1-APEX2 and Myc-CLPTM1. The relevant proteins were exogenously expressed and immunoprecipitated with anti-FLAG antibodies and revealed by immunoblotting with anti-CLPTM1 or anti-FLAG antibodies. The immunoblotting results showed that Myc-CLPTM1 co-immunoprecipitates with both F-GPER1 and its APEX2 fusion protein (Figure 7). In keeping with the nature of the proteomics results, note that this direct or indirect interaction could be demonstrated independently of stimulation with a GPER1 agonist. While we could not look at the interactions of F-GPER1 with the two α-glucosidase II subunits PRKCSH and GANAB [62] for technical reasons (incompatible tags/antibodies), additional co-IP experiments confirmed that F-PRKCSH and GANAB are indeed part of the same complex, which can form without additional exogenous Myc-CLPTM1 (Appendix A). Moreover, reciprocal co-IP experiments revealed that Myc-CLPTM1 is associated with F-PRKCSH (Appendix A). It seems likely that PRKCSH, GANAB, and CLPTM1 form a ternary complex, but confirmation will require additional experimental evidence. Based on the data of the pull-down LC-MS/MS and the co-IP with CLPTM1, it is tempting to speculate that this putative ternary complex might also associate with GPER1.

### 3.7. F-GPER1, CLPTM1, and F-PRKCSH Colocalization and Translocation

We performed transient transfection experiments with HeLa cells to coexpress various proteins. By IF, we meant to evaluate to what extent we could support our biochemical results by demonstrating colocalization, and whether any of the proteins might affect the localization of another one (Figure 8). The subcellular localization patterns of exogenously expressed tagged GPER1 and F-PRKCSH and endogenous CLPTM1, a multi-transmembrane protein, are compatible with the expected localization to the endoplasmic reticulum, and there, presumably the membrane. In agreement with our results mentioned above, exogenous GPER1, either with a 3xFLAG or HA tag (F-GPER1 and HA-GPER1, respectively), and endogenous CLPTM1 extensively colocalize, whereas exogenous F-PRKCSH at least partially colocalizes with CLPTM1.

Next, we focused on changes in localization that CLPTM1 knockdown or overexpression of one of the interaction partners might have. The shRNA-mediated knockdown of endogenous CLPTM1 drastically reduced the IF staining of CLPTM1 and incidentally also validated the use of this particular anti-CLPTM1 antibody (Figure 9, rows with shCLPTM1). Overexpression of Myc-CLPTM1 had a drastic impact on the localization of F-PRKCSH and F-GPER1, which relocalized to the nuclei of cells that seemed to have reduced DAPI staining, possibly indicating that coexpression and/or nuclear localization of F-PRKCSH or F-GPER1 with Myc-CLPTM1 induced apoptosis. The overexpression of F-PRKCSH had a similar impact on cells overexpressing HA-GPER1, whereas the knockdown of CLPTM1 did not affect the default localization of F-PRKCSH, but also induced the nuclear localization of F-GPER1 at least in a subset of cells. This unexpected impact of CLPTM1 on the localization of GPER1 could be confirmed with F-GPER1-APEX2 (Appendix A). Taken together, these results demonstrate that the default subcellular localizations of F-GPER1 and F-PRKCSH are sensitive to the levels of CLPTM1, both too little and too much. CLPTM1 is known to inhibit the GABA_A_ receptor by trapping it in the endoplasmic reticulum [40], but other than that, too little is known about it to speculate about how it might affect the localization and perhaps function of GPER1. Whether endogenous proteins are similarly affected and how CLPTM1 might regulate GPER1 functions are questions for future studies.

### 3.8. F-GPER1 Interacts with mCherry-STIM1 in HEK293T Cells

Our proteomic findings suggested that GPER1 and STIM1 could be part of the same complex, interacting directly or indirectly. We performed co-IPs of F-GPER1 and mCherry-STIM1 to confirm the MS data. We exogenously expressed F-GPER1 and F-GPER1-APEX2 and mCherry-fused STIM1 in HEK293T cells and then carried out an IP with an anti-FLAG antibody or reciprocal IPs with anti-STIM1 or anti-mCherry antibodies. Immunoblotting for the corresponding protein partner revealed a specific interaction of mCherry-STIM1 with F-GPER1 with and without APEX2 (Figure 10A). To define the domain of STIM1 required for its interaction with GPER1, we used mCherry-STIM1 fusion proteins to compare full-length STIM1 to the two STIM1 truncation mutants retaining only the N-terminal 154 or 241 amino acids. Although this co-IP experiment with an anti-FLAG antibody shows some background in the anti-mCherry blot, samples with F-GPER1 gave stronger bands for the mCherry fusions of full-length STIM1 and STIM1 truncated after amino acid 241. This indicated that the interaction of GPER1 with STIM1 requires a STIM1 region encompassing amino acids 155–241, which contain the transmembrane domain (Figure 10B).

### 3.9. mCherry-STIM1 and F-GPER1 Affect the Expression Levels of Each Other

An unexpected finding of our experiments involving co-overexpression of F-GPER1 and mCherry-STIM1 in HEK293T cells for IPs is that the expression levels of F-GPER1 and mCherry-STIM1 affect each other in a way that appears to depend on their ability to interact. Specifically, mCherry-STIM1 overexpression stabilizes F-GPER1 as well as F-GPER1-APEX2, whereas F-GPER1 or F-GPER1-APEX2 overexpression destabilizes mCherry-STIM1. The accumulation of the unrelated control protein GAPDH is not affected (Figure 10A). Furthermore, the impact of F-GPER1 on the accumulation of the two STIM1 truncation mutants (Figure 10B) suggests that the ability to interact with GPER1 may not be required for this effect and that the interplay may be indirect. It will be interesting to see whether this is also relevant for the endogenous proteins.

### 3.10. F-GPER1 and mCherry-STIM1 Colocalization and Translocation

IF imaging of F-GPER1 and mCherry-STIM1, exogenously expressed in HeLa cells, indicates that they may both be primarily localized in the membrane of the endoplasmic reticulum (Figure 11). Upon treating the cells with 1 µM of the ionophore thapsigargin (TG) for 15 min, it seems that F-GPER1 is still associated with the endoplasmic reticulum, whereas mCherry-STIM1 adopts a much more punctiform pattern (Figure 11, second row). This recapitulates what was already known about STIM1, namely that it accumulates in puncta located in the endoplasmic reticulum and at endoplasmic reticulum–plasma membrane junctions upon calcium store depletion [63,64]. When F-GPER1 and mCherry-STIM1 were coexpressed, they colocalized but seemed to retain the same subcellular localization. In contrast, upon TG treatment, they either colocalized in a perinuclear or, in the case of F-GPER1, even in a nuclear region (Figure 11, bottom rows). As mentioned above for the impact of CLPTM1 on the localization of F-GPER1, here again, it seemed that cells with coexpression of F-GPER1 and mCherry-STIM1 stained more weakly with DAPI, and therefore potentially were in the early stages of apoptosis. Collectively, the biochemical and IF experiments strongly support the conclusion that STIM1 interacts as part of the same complex and that there may be a functional interaction as well.

## 4. Discussion

Despite numerous studies on GPER1 over the last few years, it remains controversial whether GPER1, as a nonclassical estrogen receptor, can indeed bind to and mediate nongenomic and rapid estrogen responses [18,19,24]. In parallel, there is an ever-increasing interest in GPER1 as a target in the context of several cancers, particularly breast cancer [9,12,13,15,16,17]. We, therefore, decided to improve our knowledge of the GPER1 interactome, hoping that this could ultimately help decipher its activation mechanisms and signal transduction pathways, understand its physiological and pathological roles, and promote the development of efficient treatments for diseases where GPER1 might be relevant [31,34,35,36].

An important conclusion from our investigations is that the activity assay and the GPER1 interactome are not affected by estrogen. In agreement with some previous publications, we must conclude that GPER1 has a constitutive, ligand-independent activity [24,25,29]. Formally, we cannot exclude that much higher E2 concentrations than the ones we have used (100 nM) would elicit responses beyond the constitutive ones that we have observed, but responses at pharmacological doses even further above the highest concentrations of about 1.5 nM during the menstrual cycle are unlikely to be physiologically relevant. Although we have performed almost all of our experiments with GPER1 exogenously expressed in HEK293T cells, we speculate that this may apply to a variety of other biological contexts. Moreover, we cannot exclude that some downstream signaling events of GPER1, other than Rac1 signaling, may display a different ligand dependence, as previously reported for other GPCRs [65].

Our proteomic results reveal a significant enrichment of proteins of the membrane of the endoplasmic reticulum, the endoplasmic reticulum–Golgi intermediate compartment, the nuclear envelope, and the nucleus rather than proteins of the plasma membrane. Our results support the notion that GPER1 preferentially localizes to intracellular membranes and the nucleus, rather than the plasma membrane [3,56,66,67]. Interestingly, a study of the interactome of the angiotensin II type 1 receptor (AT1R) with APEX2 revealed a significant enrichment of different variants of G_α_ subunits of heterotrimeric G proteins in the absence of angiotensin [31]. The fact that G_αs_ subunits were not part of our GPER1 interactome supports the idea that at least some of the constitutive activity of GPER1 may mirror the dissociation of G_αs_ from the βγ heterodimer upon activation of GPCRs [68].

Previous reports established that GPER1 can also localize to the nucleus in an importin-dependent manner [56,66,67]. The translocation of the N-glycosylated heptahelical receptors from the endoplasmic reticulum to the nucleus can be promoted by an NLS in a Ran-GTP/importin-dependent manner or without NLS in a manner dependent on other transport proteins [69,70]. In this context, it is interesting that our proximity labeling–LC-MS/MS analysis showed enrichment of NUP210 and torsins, which are associated with the membranes of the endoplasmic reticulum and the nuclear envelope [57]. It might be worthwhile to determine experimentally whether these proteins are involved in the nuclear localization of GPER1.

Using two very different methods to identify potential GPER1 interactors, we were able to identify 73 high-confidence hits, of which we further validated several. Intriguingly, despite our stringent approach, we could not identify any of the previously reported GPER1 interactors. In fact, this was also the case when we considered the data from the two approaches separately. At this point, we can only assume that at least one critical experimental parameter was different in our experiments compared to those previously described. Ultimately, only experimental scrutiny can sort this out and generate a comprehensive and validated list of GPER1 interactors that are also functionally relevant.

Of all the potentially interesting GPER1 interactors revealed by our proteomic results, we decided to focus on CLPTM1, PRKCSH, GANAB, and STIM1. CLPTM1 is a multipass transmembrane protein of the membrane of the endoplasmic reticulum. It is known as a negative regulator of the GABA_A_ receptor; CLPTM1 exerts this effect by trapping the GABA_A_ receptor in the endoplasmic reticulum and limiting its trafficking toward the plasma membrane [40,71]. Using imaging and IPs, we were able to confirm the proteomic results indicating that GPER1 directly or indirectly interacts with CLPTM1. Our IF results revealed that the interaction between F-GPER1 and CLPTM1 affects F-GPER1 localization. Upon overexpression of both Myc-CLPTM1 and F-GPER1, both proteins appear to relocalize to a perinuclear and/or nuclear region. Where exactly the two proteins become trapped, why this happens, how, and whether this might explain the apparent cellular toxicity of co-overexpression remain to be determined. Interestingly, the CLPTM1 knockdown resulted in nuclear localization of F-GPER1 and F-GPER1-APEX2, suggesting that the proper localization of GPER1 may require a finely balanced ratio to its interaction partner CLPTM1.

Our proteomic results showed a highly significant enrichment of proteins related to the maturation and glycosylation of proteins, including PRKCSH, GANAB, UGGTI, MOGS, CANX, CALR, and ERP29. They all play a role in the calnexin cycle [72]. PRKCSH and GANAB are the regulatory and catalytic subunits of α-glucosidase II, respectively. They are involved in trimming N-glycan groups and also in quality control of glycoproteins in the lumen of the endoplasmic reticulum as part of the calnexin cycle, which removes the innermost α-1,3-linked glucose residue from Glc2Man9GlcNAc2. Co-overexpression of F-PRKCSH and HA-GPER1 caused the relocalization of HA-GPER1 to the nucleus and/or the perinuclear region. Similarly, upon overexpression of Myc-CLPTM1, F-PRKCSH became associated with the nucleus. Full-length PRKCSH has been shown to localize in the endoplasmic reticulum, whereas a truncated form in breast cancer cells could be seen associated with the nucleus [73], and even the plasma membrane, and intracellular vesicles [74]. Note that we did not evaluate the impact of GPER1 overexpression on the localization of PRKCSH, but it has been reported for breast cancer cells that PRKCSH can translocate to the nucleus in a complex with the fibroblast growth factor 1 and its receptor FGFR. It will be interesting to find out how PRKCSH can act to promote the nuclear localization of GPER1. At this point, it is unknown whether the previously reported nuclear localization of GPER1 upon interfering with glycosylation with tunicamycin or genetically with a point mutation in GPER1 [56,66] involves PRKCSH and GANAB as subunits of α-glucosidase II. It is tempting to speculate, based on these results and our additional co-IP experiments, that CLPTM1 may be involved in the calnexin cycle, affecting the processing, maturation, and trafficking of glycoproteins, including GPCRs, through its interaction with the α-glucosidase II subunits PRKCSH and GANAB.

STIM1 is yet another transmembrane protein of the endoplasmic reticulum that we have demonstrated to interact, directly or indirectly, with GPER1. Our co-IP experiments indicated that the interaction requires the first 241 amino acids of STIM1, which correspond to its transmembrane and luminal domains. STIM1 is known as a Ca^2+^ sensor in the endoplasmic reticulum, where it plays a crucial role in the activation of store-operated Ca^2+^ entry (SOCE) and calcium signaling [75,76]. Upon depletion of Ca^2+^ in the endoplasmic reticulum, STIM1 is activated, which involves a conformational change promoting its oligomerization and augmentation of SOCE [77]. As for the other GPER1 interactors discussed above, mCherry-STIM1 and F-GPER1 appear to affect each other’s subcellular localization. As mentioned above, we found that F-GPER1, overexpressed by itself, is associated with cellular structures consistent with a localization to the membrane of the endoplasmic reticulum. mCherry-STIM1 appears to localize to similar structures, even though there may be more peripheral localizations as well. Upon exposure of cells to the ionophore TG, which would trigger the activation of endogenous and presumably also exogenous STIM1, F-GPER1 may become more diffusely cytoplasmic (Figure 11), whereas overexpressed mCherry-STIM1 itself becomes more concentrated at puncta, which are structures known to be associated with the activation of SOCE [78]. When overexpressed together, F-GPER1 and mCherry-STIM1 may drive each other to a more perinuclear localization. In the presence of TG, both proteins appear to become more perinuclear, with F-GPER1 potentially becoming even nuclear, and overexpression of F-GPER1 diminishes puncta formation by mCherry-STIM1, thus potentially diminishing STIM1 activation. In this context, it is important to remember that we had found an unexpected impact of F-GPER1 overexpression on exogenously expressed mCherry-STIM1 in that mCherry-STIM1 protein levels are dramatically reduced. Whether these two phenomena are related will require additional studies. Suffice it to say for now that nuclear localization in a number of scenarios presented above was associated with reduced DAPI staining of the nuclei. This potentially indicates that an excessive number of complexes of GPER1 with some of its interactors may be toxic to cells and induce apoptosis, which may relate to the enrichment of proteins associated with the GO terms “unfolded protein response” and “ERAD pathway”. Under physiological conditions, it is conceivable that GPER1 plays a role in fine-tuning the activity of various interactors. In the case of STIM1, this would be expected to affect calcium homeostasis. Intriguingly, GPER1 has very recently been linked to SOCE, although SOCE was shown to be reduced by the GPER1 agonist G-1 [79]. Based on our results, we propose that GPER1 may also affect SOCE in a manner that is constitutive or regulated by yet other signals.

## 5. Conclusions

Our two-pronged proteomic approach to elucidating GPER1 interactors yielded a promising list of candidates. Somewhat surprisingly, considering part of the GPER1 literature, we were unable to find E2-induced GPER1 interactions. However, our additional experimental validation of several hits further confirmed the constitutive nature of these interactions. In our hands, this correlates with the constitutive GPER1 activity in a functional assay. A clear limitation of our results is that we overexpressed the proteins of interest to be able to use tagged or fluorescent versions for easier detection. Moreover, all results presented here were obtained with the GPER1-negative cell lines HEK293T and HeLa. It is fair to speculate that many of our results will be portable to other cell types with endogenously expressed GPER1, at the very least under some cellular conditions. We expect that extending our studies to the other hits of our preliminary list of GPER1 interactors will contribute to sorting out the ongoing controversy about GPER1 agonists and perhaps even lead to the discovery of yet other physiological agonists and antagonists.

## Figures and Tables

**Figure 1 cells-12-02571-f001:**
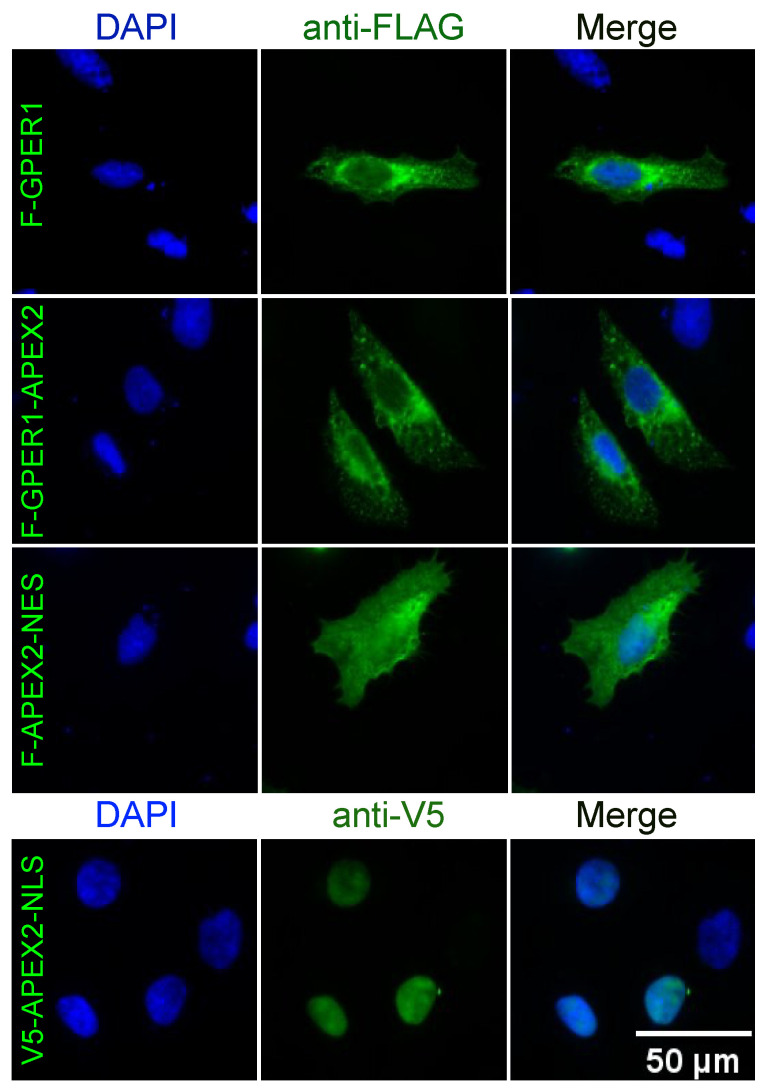
Subcellular localization of F-GPER1 and APEX2 fusion proteins in HeLa cells. As a control experiment, the indicated proteins were transiently expressed in HeLa cells and immunostained with antibodies specific for their respective tags. DAPI was used to stain the nuclei. NES and NLS, nuclear export and nuclear localization signals, respectively. See Appendix A for schemes of plasmids/proteins. Images were captured with a fluorescence microscope (Zeiss). Scale bar = 50 μM.

**Figure 2 cells-12-02571-f002:**
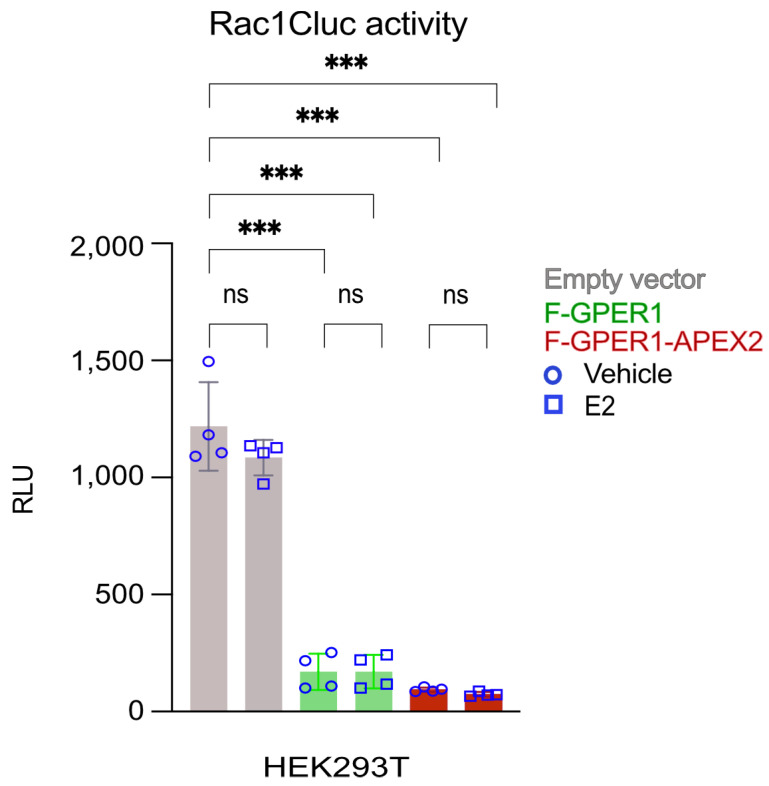
Conformational Rac1 sensor assay indicates constitutive GPER1 activity. HEK293T cells were transiently cotransfected with the Rac1 sensor plasmid Rac1Cluc and expression plasmids as indicated and treated with 100 nM E2 as shown. The bar graph shows relative luminescence units (RLU) of the Rac1Cluc split luciferase, each bar representing the average of 4 data points (shown as circles and squares for vehicle and E2 treatments, respectively) of the 10 min time points of two biologically independent experiments (with two replicates each). The statistical analysis was performed with a one-way ANOVA test. ns, statistically non-significant difference; ***, *p*-values of <0.0001.

**Figure 3 cells-12-02571-f003:**
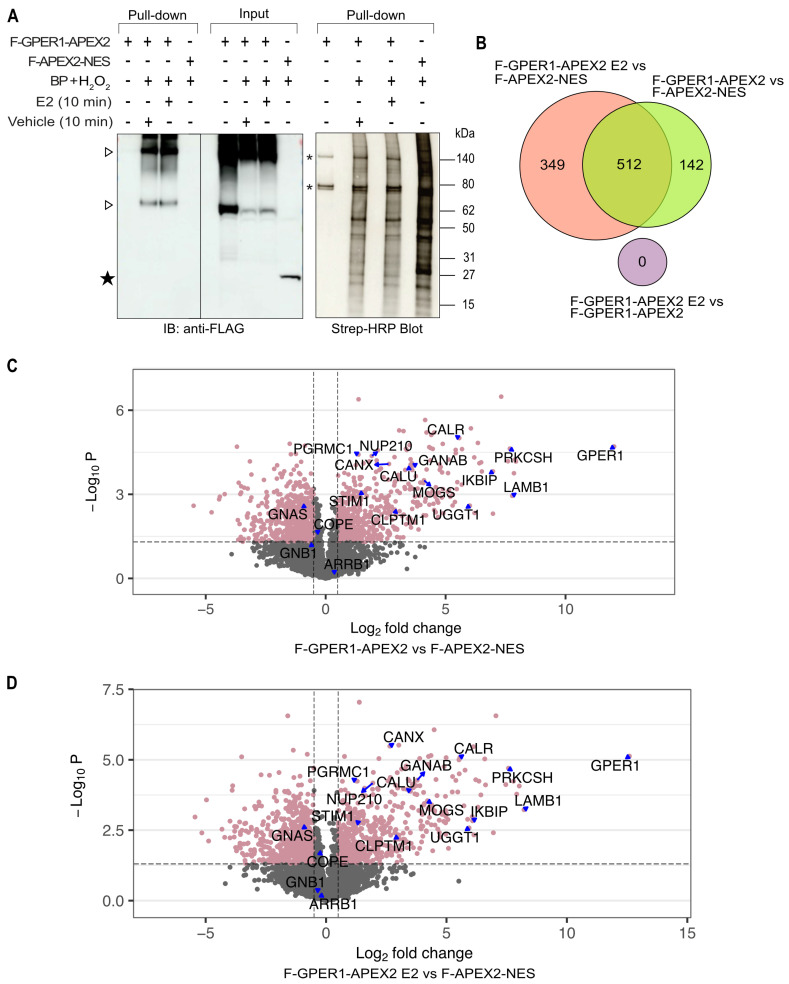
APEX2-mediated proximity labeling experiment and analysis of LC-MS/MS results. (**A**) Quality control experiment for APEX2-mediated biotinylation and pull-down with streptavidin beads. The left part shows anti-FLAG immunoblots (IBs) of the input and pulled-down material, and the right part the pulled-down material displayed with streptavidin-HRP. Biotinylation was triggered by the addition of BP and H_2_O_2_ to HEK293T cells transiently expressing the indicated APEX2 fusion proteins. ▷, bands of about 70 and 140 kDa corresponding to the monomeric size of F-GPER1-APEX2 and its glycosylated and/or dimeric/oligomeric forms, respectively. ★, band corresponding to F-APEX2-NES. *, bands corresponding to major endogenously biotinylated proteins (independently of APEX2). (**B**) Venn diagram of the data of the proximity labeling LC-MS/MS experiment. It shows the number of proteins that differ for the indicated binary comparisons, and, in the overlaps of the circles, for the comparisons of the respective binary comparisons. Note that there were no proteins with significant differences between F-GPER1-APEX2 treated with E2 for 10 min and untreated F-GPER1-APEX2, and none with the other two comparisons (symbolized by the offset small purple circle). The sizes of the circles only approximately reflect the number of proteins. Cutoff values: *p*-value ≤ 0.05, and *q*-value ≤ 0.05. (**C**,**D**) Volcano plots of the data obtained with F-GPER1-APEX2 without (panel C) and with E2 (panel D) versus F-APEX2-NES. The proteins represented by pinkish mauve-colored dots (with some individual proteins pointed out by blue-filled triangles) correspond to the most stringent cut-off: *p*-values ≤ 0.05 (horizontal hashed line) and *q*-values ≤ 0.05, and log2 fold change ≥0.5 or ≤−0.5 (vertical hashed lines), corresponding to enriched or depleted proteins, respectively.

**Figure 4 cells-12-02571-f004:**
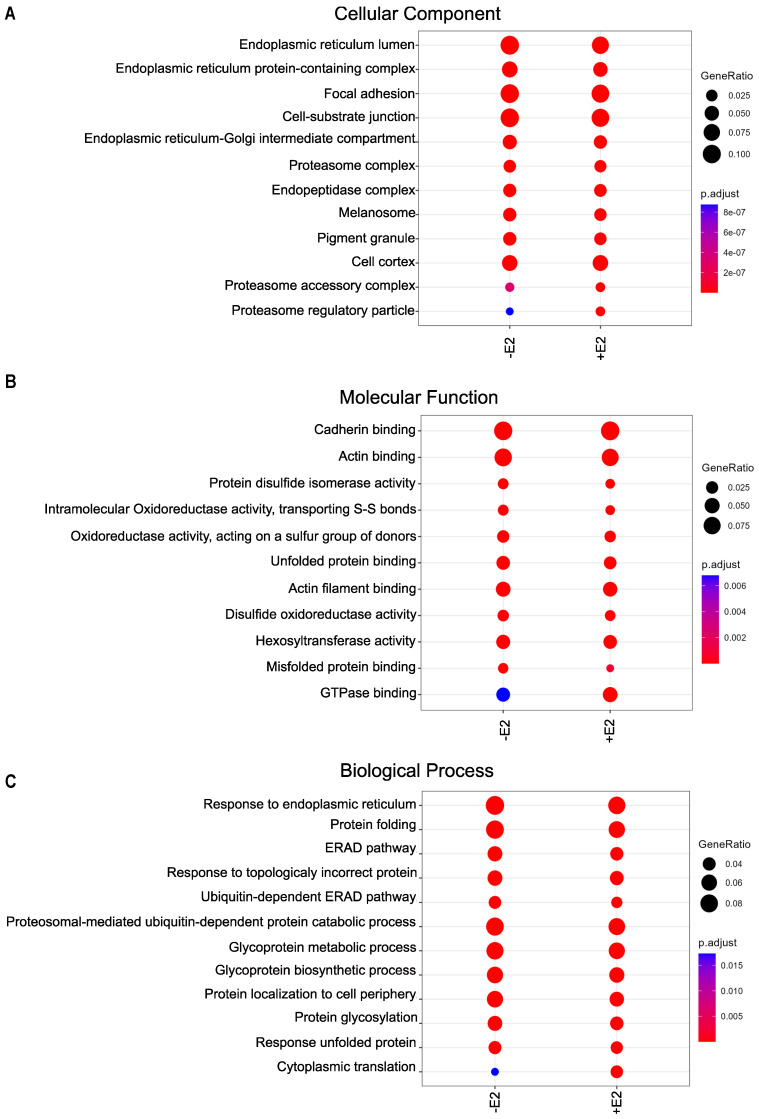
GO analysis of the proximity labeling–LC-MS/MS data. GO term enrichment analyses of terms associated with “Cellular Component” (**A**), “Molecular Function” (**B**), and “Biological Process” (**C**). The color scale on the right indicates *q*-values (adjusted *p*-values). GeneRatio (and the size of the colored dots) indicates the fraction of enriched proteins relative to all proteins associated with the indicated GO term; ERAD, endoplasmic reticulum-associated degradation.

**Figure 5 cells-12-02571-f005:**
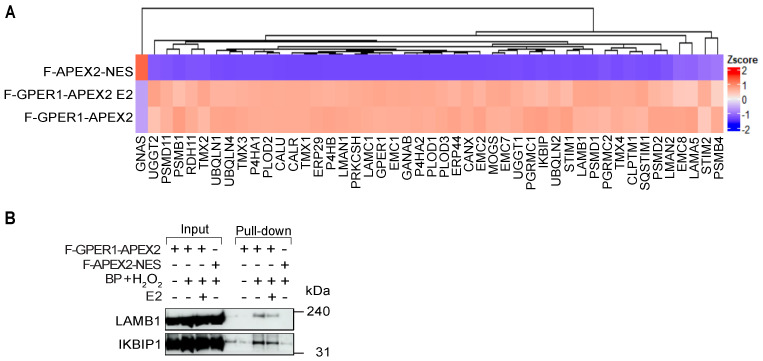
Enrichment pattern of some of the proximity labeling–LC-MS/MS data and validation of IKBIP and LAMB1 as GPER1-proximal proteins. (**A**) Hierarchically clustered heat map illustrating the enrichment of certain biotinylated proteins in the presence of the APEX2 fusion proteins and E2 as indicated on the left. For this illustration, only proteins corresponding to some of the highlighted GO terms (specifically: ERAD pathway, proteasome complex, protein glycosylation, cadherin binding, focal adhesion, endoplasmic reticulum–Golgi intermediate compartment) were selected. The heat map was generated with the average Log_2_ LFQ intensities of 3 biological replicates. (**B**) Immunoblot analysis of a pull-down of biotinylated proteins for the enriched presence of IKBIP and LAMB1 upon proximity labeling mediated by F-GPER1-APEX2 with and without E2 compared to the negative control protein F-APEX2-NES. LAMB1 and IKBIP1 were revealed with their respective specific antisera. Numbers on the right point out molecular weights of marker proteins close by.

**Figure 6 cells-12-02571-f006:**
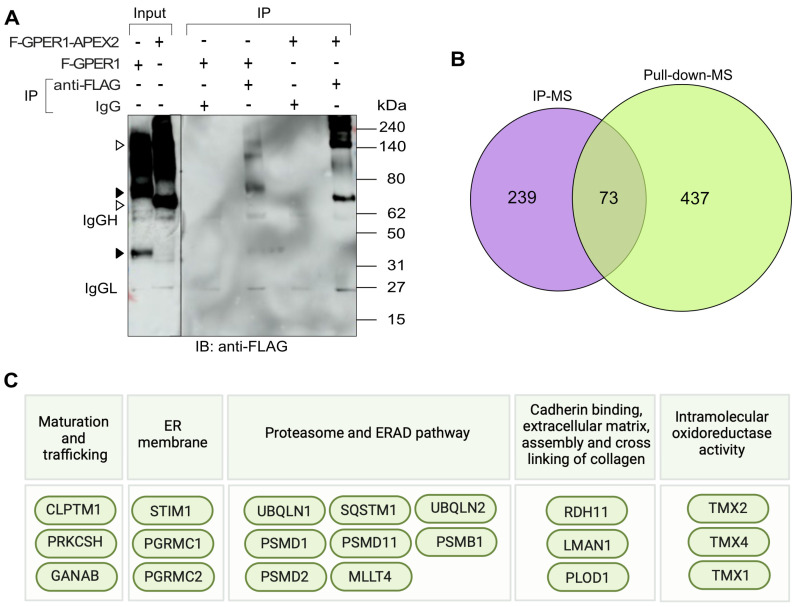
IP-LC-MS/MS experiment and data analysis. (**A**) Quality control experiment for the anti-FLAG IP of proteins associated with F-GPER1-APEX2 and F-GPER1, transiently expressed in HEK293T cells. Mouse IgG antibodies were used as negative control. The open and closed arrowheads on the left point to the bands corresponding to the proteins F-GPER1-APEX2 (see also Figure 3A) and F-GPER1 (as for F-GPER1-APEX2, representing glycosylated forms and/or dimers or hetero-oligomers), respectively. The positions of the heavy (IgGH) and light (IgGL) chains of the antibodies are indicated. (**B**) Venn diagram of top hits of the IP-LC-MS/MS and proximity labeling–LC-MS/MS experiments. The top hits of the proximity labeling–LC-MS/MS experiment (selected with the same cutoff as for Figure 3B, and essentially corresponding to the ones common to F-GPER-APEX2 without and with E2) were compared to those of the IP-LC-MS/MS experiments (the ones common to F-GPER1 and F-GPER1-APEX2, sorted based on iBAQ score). The sizes of the circles only approximately reflect the number of proteins. (**C**) Selection of some of the 73 common proteins of panel B in relationship to the GO terms they are associated with. The illustration was created with BioRender.com. ER, endoplasmic reticulum.

**Figure 7 cells-12-02571-f007:**
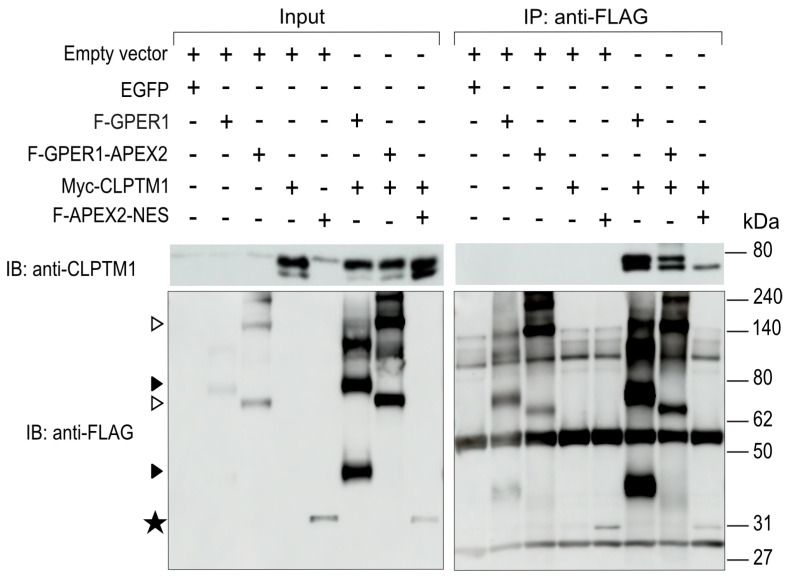
GPER1 interacts with CLPTM1. Co-IP experiment with HEK293T cells transiently expressing indicated proteins. The co-IP was performed with an anti-FLAG antibody, and the immunoblots (IBs) with anti-CLPTM1 or anti-FLAG antibodies. F-APEX2-NES served as negative control. In all cases, two expression vectors were transfected (in some cases one was the empty expression vector), and at least one protein was exogenously expressed (which in some cases was EGFP as negative control). The open and closed arrowheads on the left point to the bands corresponding to the various forms of the proteins F-GPER1-APEX2 and F-GPER1, respectively (see also Figure 3A and Figure 6A). ★, band corresponding to F-APEX2-NES.

**Figure 8 cells-12-02571-f008:**
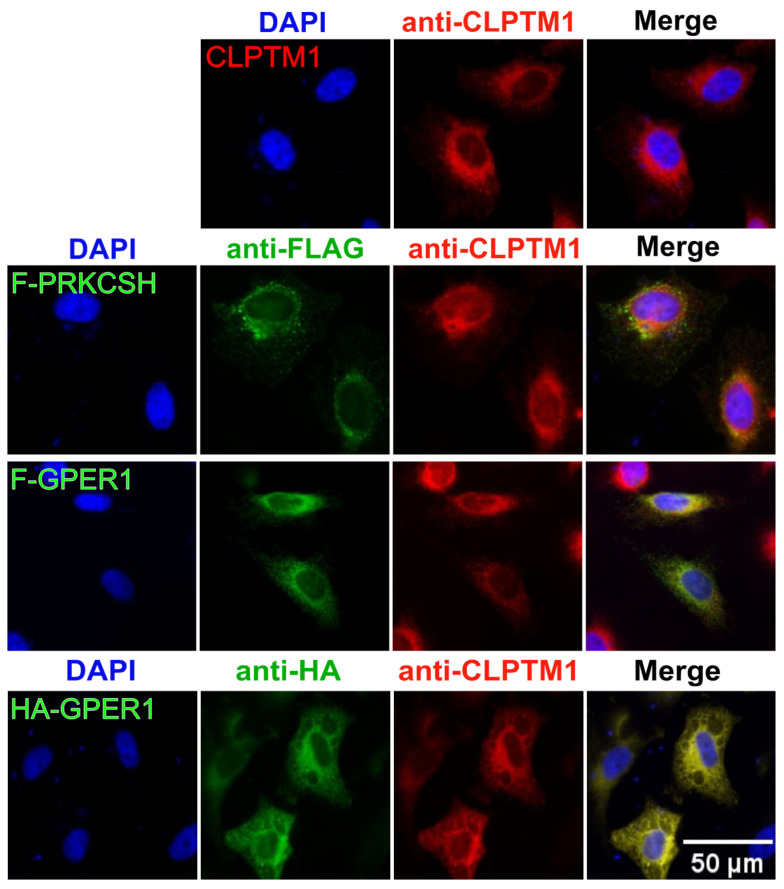
Subcellular localization of CLPTM1, PRKCSH, and GPER1. IF experiment with HeLa cells that were transiently transfected to express Myc-CLPTM1, F-PRKCSH, F-GPER1, and HA-GPER1 separately. Only the top row of images shows cells expressing both exogenous Myc-CLPTM1 and endogenous CLPTM1, both recognized by the same antibody; for all others, it is only endogenous CLPTM1. Immunostaining and DAPI staining as mentioned for Figure 1. Scale bar = 50 μM.

**Figure 9 cells-12-02571-f009:**
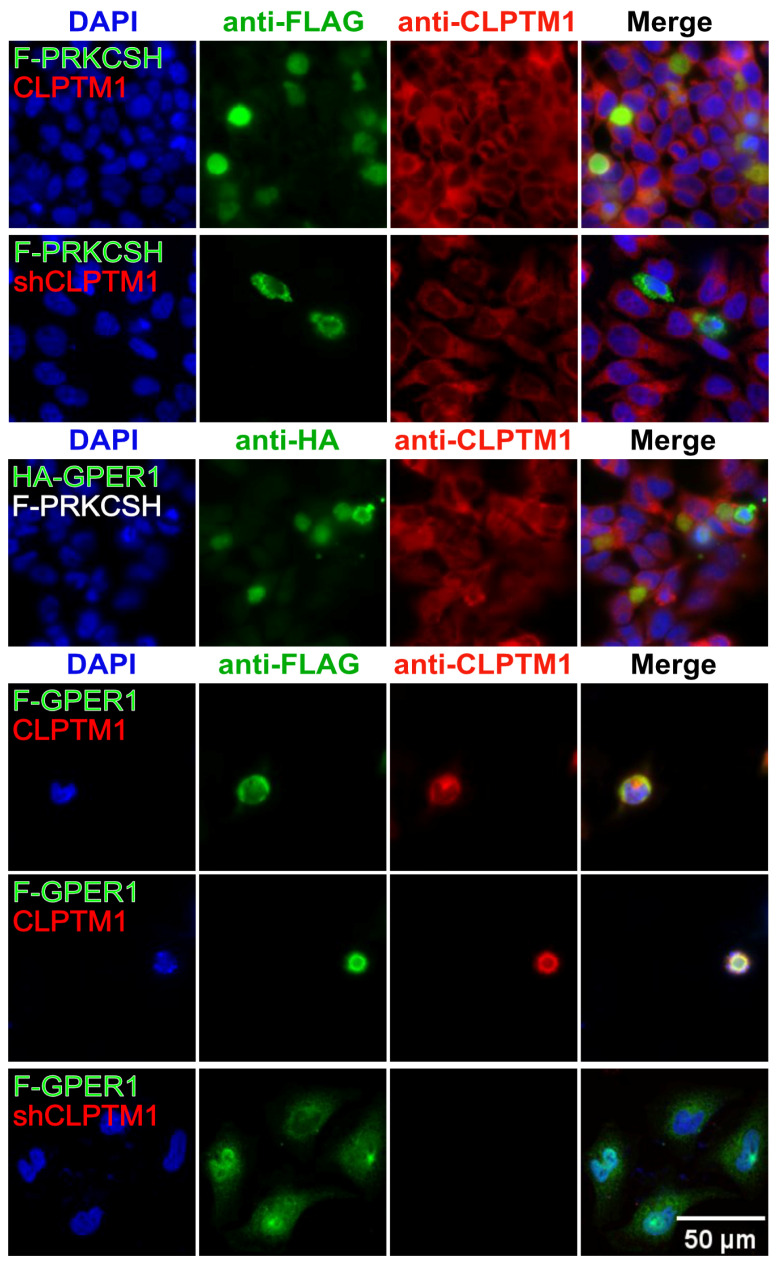
Subcellular localization of GPER1, CLPTM1, and PRKCSH in combination. IF experiment with HeLa cells that were transiently transfected with the indicated plasmids (always mentioned in the first and DAPI-stained image of each row on the far **left**). shCLPTM1 allows the production of shRNA to knock down endogenous CLPTM1 expression. Note that CLPTM1 written in red indicates exogenous Myc-CLPTM1 and that F-PRKCSH written in white indicates that it was exogenously expressed but not stained for. Moreover, note that there are two rows for the coexpression of F-GPER1 and Myc-CLPTM1. Scale bar = 50 μM.

**Figure 10 cells-12-02571-f010:**
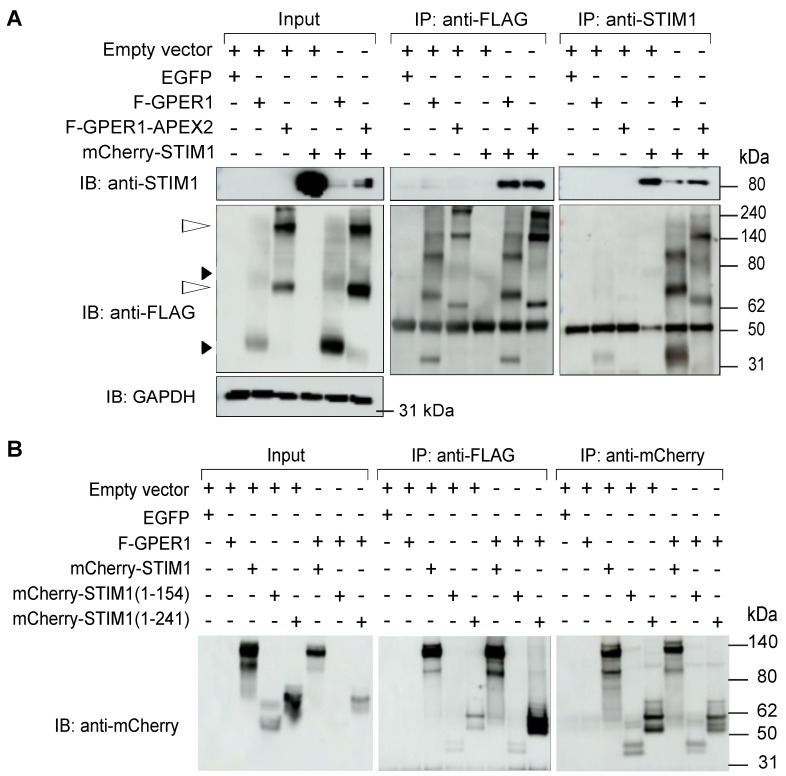
GPER1 interacts with STIM1. (**A**) Co-IP experiment with HEK293T cells transiently expressing indicated proteins demonstrating that mCherry-STIM1 interacts with both F-GPER1 and F-GPER1-APEX2. The experiment was set up in an analogous way to the co-IP experiment of Figure 7. GAPDH was used as a loading control for the input samples. The open and closed arrowheads on the left point to the bands corresponding to the proteins F-GPER1-APEX2 and F-GPER1, respectively (representing glycosylated forms and/or dimers or hetero-oligomers; see also Figure 6). (**B**) Co-IP experiment with STIM1 truncation mutants to map STIM1 domain required for interaction.

**Figure 11 cells-12-02571-f011:**
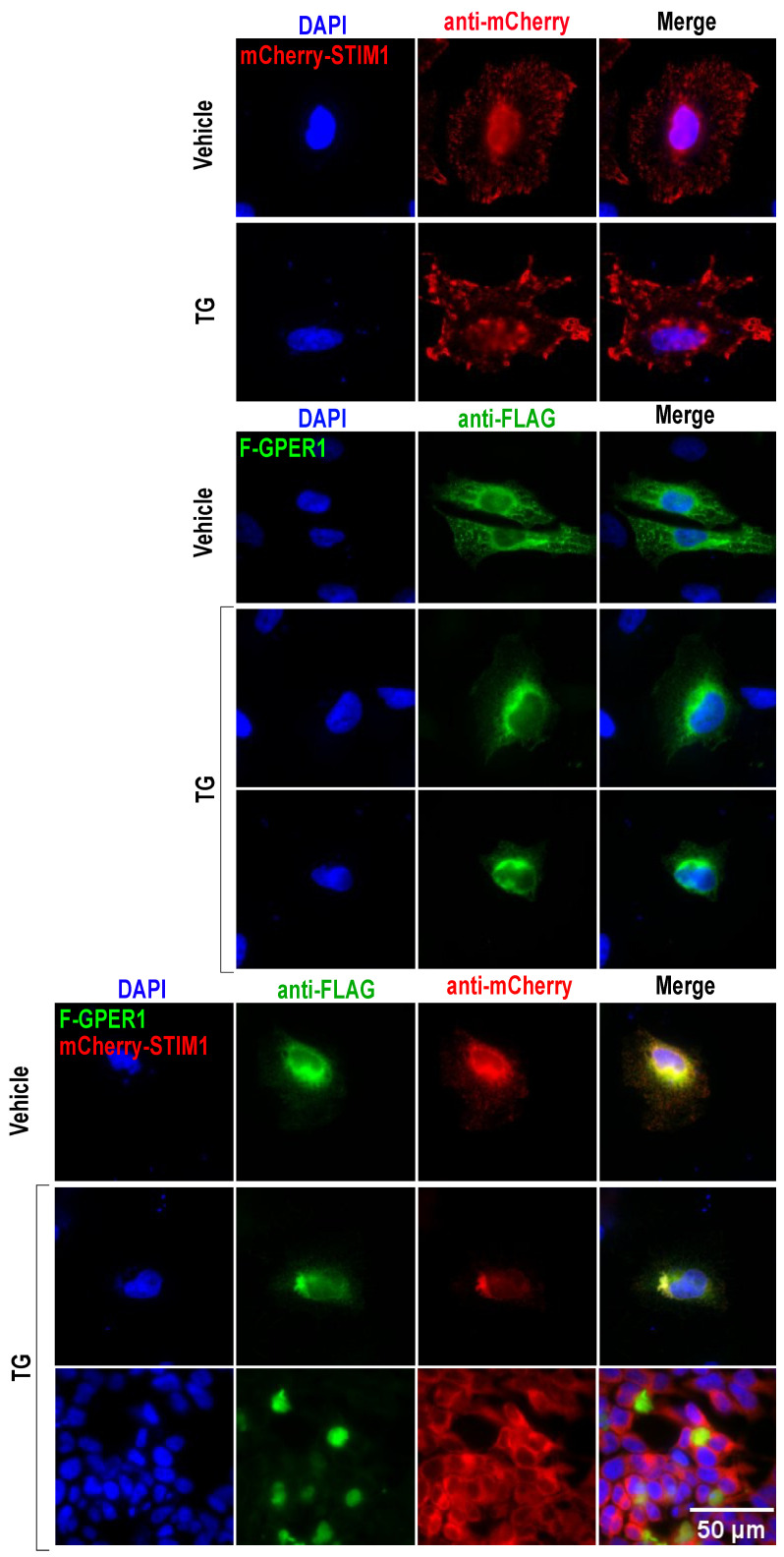
Subcellular localization of GPER1 and STIM1 and impact of the ionophore thapsigargin. IF experiment with HeLa cells with the indicated exogenously expressed proteins and immunostaining with indicated antibodies. TG, treatment of cells with 1 µM thapsigargin for 15 min prior to fixation. Scale bar = 50 μM.

## Data Availability

The LC-MS/MS proteomics data have been deposited to the ProteomeXchange Consortium via the PRIDE partner repository (https://www.ebi.ac.uk/pride (accessed on 13 October 2023)) with the dataset identifier PXD041943, and for a subset, they are available as Appendix A.

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
