# Peer review of "Proteomic Analyses of the G Protein-Coupled Estrogen Receptor GPER1 Reveal Constitutive Links to Endoplasmic Reticulum, Glycosylation, Trafficking, and Calcium Signaling"

_cells, 2023, doi:10.3390/cells12212571_

Round 1

Reviewer 1 Report

Comments and Suggestions for Authors

The authors examined steroid hormone 17β-estradiol (E2)-induced activity of G protein-coupled estrogen receptor 1 (GPER1) in an in vitro Rac1 sensor assay and novel interaction partners of GPER1 by ascorbate peroxidase 2 (APEX2)-mediated proximity labelling and immunoprecipitation followed by liquid chromatography-tandem mass spectroscopy (LC-MS/MS) and pull-down analyses. They found E2-independent constitutive activity of GPER1 and 73 novel potential GPER1 interactors including cleft lip and palate transmembrane protein 1 (CLPTM1), protein kinase C substrate 80K-H (PRKCSH, glucosidase II subunit beta, regulatory subunit), neutral α-glucosidase AB (GANAB, catalytic subunit of glucosidase II), and stromal interaction molecule 1 (STIM1, Ca2+ sensor). The manuscript looked well written and the experiments were well designed except for the ligand concentrations. However, there are some unclear points, some of which would originate from too low ligand concentration, or typos. Please address the following points.

Major points:

1)    A paper (Harding AT, Goff MA, Froggatt HM, Lim JK, Heaton NS. GPER1 is required to protect fetal health from maternal inflammation. Science. 2021, 371(6526), 271–276. doi: 10.1126/science.aba9001) has reported 20-µM E2-activated GPER1-mediated suppression of type I interferon signaling. They examined responses to 500 nM E2, resulted in no responses, whereas 5-µM and 20-µM E2 or 5-µM G1 induced small and robust responses, respectively. The E2 concentration of 100 nM which was used in the present study would be too low to conclude the ineffectiveness of E2 as the agonist of GPER1, won’t it?

2)    In the above paper, GPER1, PSMA5, CPEB4, EIF2AK1, TRMT1L, and UTP14A were reported as candidate interferon signaling regulators. In some of red dots of Figure 3C and D in the present study, the authors might identify some of PSMA5, CPEB4, EIF2AK1, TRMT1L, and UTP14A as GPER1 interaction partners for potential cross talk with the type I interferon signaling pathway in the innate immune response? Please describe them in Results or Discussion with the above reference as possible functions and cross talk signaling pathways of GPER1.

3)    Please clearly describe the principle of Rac1 sensor assay and the reference (#24?) in Materials and Methods and provide a schematic diagram in Figure 2 and add a schematic representation of Rac1 plasmid in Figure S1 for better readability. How can the CLuc luminescence intensity of Rac1CLuc proportionate changes in the activity of GPER1 or GPER1-mediated responses? 

4)    In the section 2.4, please describe the number of wells for each data point of Rac1CLuc activity.

5)    In Figure 2, please add gray and green bars for Empty vector (only Rac1CLuc?) and F-GPER1 with Rac1CLuc?.

6)    Please clearly describe the principle of APEX2-mediated proximity labelling assay with biotin and the references (#31–33?) in Materials and Methods and provide a schematic diagram with biotin in Figure 3 for better readability. It could find interaction partners in the mitochondrial matrix and nucleus? 

7)    Please correct the Venn diagram in Figure 3B. I could not find any overlaps between circles and/or more than six numbers of labeled proteins for understanding what the authors would like to demonstrate in a single reddish pink circle (no outline).

8)    In Figure 3C–D, please improve the readability of abbreviated protein names by moving them from dotted areas to non-dot areas on the respective sides in a manner of the same order of the vertical and horizontal locations as in the dots and an enhancement of top four hits of PRKCSH, CLPTM1, GANAB, and STIM1 and their dots or pointing lines or highlighted backgrounds by different colors. 

9)    In Figure 4, the data indicate that E2 induced significant associations of GPER1 with proteasome regulatory particle, GTPase binding, and cytoplasmic translation compared to those of no applied E2 condition. Are these correct? Why dd not the authors describe them? 

10) In Figure 5A, the heatmap looked inconsistent with the E2-induced significant associations of GPER1 with proteasome regulatory particle, GTPase binding, and cytoplasmic translation in Figure 4. Why?

11) In L.491, “Our findings also demonstrate significant enrichment of the nucleoporin NUP210 of the” would be “Moreover, our results indicate GPER1-mediated enrichment of the nucleoporin 210 (NUP210) as an essential trafficking regulator in the”

12) As similarly in Figure 3B, please correct the Venn diagram in Figure 6B. 

13) In Figure 8, relative sizes of the nuclei looked different between the four photos of Hela cells. Please adjust scale sizes of all photos to that of the photo (F-PRKCSH) in the 2nd row. Hela cells in the bottom row looked larger in cell sizes compared to those in the upper rows. They were sick by infection with mycoplasma or other causes?

14) In Figure 9, relative sizes of the nuclei again looked different between the seven photos of Hela cells. Please adjust scale sizes of all photos to the same one or show a scale bar in the right photo in each row. The enlargement of the middle five row photos seemed too small to observe subcellular localization of respective molecules. Photos of a single cell with a strange nucleus in the 5th and 6th row seemed insufficient as a typical result.

15) Despite the description in L.642, I could not find the C figure in Figure 10. Please add Figure 10C or correct the text.

16) TG or TG treatment is easily confused with transgenic, triglyceride, or transglutaminase and their treatments. Please replace it with 1-µM TG-treated or thapsigargin-treated.

17) In Discussion, no GPER1-mediated response to 100-nM E2 would be too weak to conclude that the activity assay and the GPER1 interactome are not affected by estrogen (L.680).

18) Based on the descriptions for the association of GPER1 with cancers in Abstract and Introduction, Discussion would be structured to possible crosstalk between the identified GPER1 interaction partners or its downstream molecules and cancer-associated signaling pathways including apoptosis disorders and innate immune responses against tumor cells such as type I interferon signaling via GPER1 with or without G15 or other types of G proteins. The authors would rearrange the structure of Discussion. In addition, please clearly discuss how the identified molecules could associate with tumor and tumor angiogenesis factors or oncometabolites or mitochondrial disorders for cancer metabolism by being switched from the energetic metabolism to anabolic metabolism for uncontrolled cell proliferation.

Minor points:

19) In Introduction, for an easy comparison with the present conclusion of E2-independent constitutive activity of GPER1, the authors would refer to the #24 reference as E2/G-1-independent activity of GPER1.

20) In Abstract, please add full spellings (Abbrev.) of APEX2, Rac1, CLPTM1, PRKCSH, GANAB, and STIM1.

21) In Introduction or Materials and Methods, please add full spellings (Abbrev.) of APEX2, PDZ, CLPTM1, PRKCSH, GANAB, STIM1, and Rac1 in the first appearance.

22) In L.73, duplicated “transfected” would be deleted.

23) Please use an en dash for numeric ranges instead of a hyphen such as in L.145, L.323, L.331, L.412, L.438, L464, and L.529.

24) In L.148 & L.175, Rac1Cluc -> Rac1CLuc.

25) In L.151, Immunofluorescence -> Immunofluorescence (IF).

26) In L.172, please describe the product name of Zeiss fluorescence microscope.

27) Please use the minus sign instead of a hyphen such as in L.209, L.296, and L.459.

28) In L.235 & L.417 & L.451, MS -> LC-MS/MS? and L.250, MS -> liquid chromatography-tandem mass spectroscopy (LC-MS/MS)?

29) In L.326, LC-MS/MS -> Trapped ion mobility spectroscopy (TIMS)? and L.334, TIMS PASEF -> TIMS parallel accumulation-serial fragmentation (PASEF)?

30) L.402, RacCLuc1 -> Rac1CLuc

31) Please unify the descriptions between the text (L.420, about 73 kDa) and the legend (L.448, about 70 kDa) and between the text (L.422, ≥140 kDa) and the legend (L.448, 140 kDa).

32) In L.431, Figure 3A -> in Figure 3A.

33) In Figure 4, -E2 -> −E2.

34) Please describe full spellings for ERAD (L.480), CANX (L.483), CALR, ERP29 (L.484) in the first appearance.

35) L.491, the nucleoporin NUP210 -> the nucleoporin 210 (NUP210)

36) L.509, IP-MS -> IP–LC-MS/MS?

37) L.516, MS -> LC-MS/MS?

38) In L.552, please add “(data not shown)”.

Comments on the Quality of English Language

Readability would be improved by corrections in several points.

Author Response

Please see separate file

Reviewer 2 Report

Comments and Suggestions for Authors

This is a high-quality study on the G-protein coupled estrogen receptor regulation of proteins. This is a study that extends our knowledge of this novel type of protein receptor.  The study design is proper as well as the presentation of results and description. Only minor concerns are listed below.

1.    In the title information that the study is performed on the cell line should be added
2.    Abbreviations should be extended when mentioned for the first time
3.    Add detailed information on the used cell line e/g. supplier, passage number, tests you used in your laboratory to routinely check morphological and biochemical characteristics of these cells
4.    Add study limitations

Author Response

Please see separate file

Round 2

Reviewer 1 Report

Comments and Suggestions for Authors

1)    As described previously in the 1st point, I have doubted whether 100-nM E2 was sufficiently high. The authors refered the results in SKBR3 cancer cells (Mol. Endocrinol. 2000, 14, 1649-1660). However, the authors should consider differences in unidentified disorders between SKBR3 cancer cells and HEK293 cells. In the reports using HEK293 cells (ref.#6: Endocrinology 2007, 148, 3236-3245 ; ref.#25:  Mol. Pharmacol. 2023, 103, 48-62), 10-nM E2 increased the cAMP level by about 10% (ref.#6) or 5-μM E2 induced no significant change in cAMP level (ref.#25). Even between GPER1 expression in HEK293, researchers reported inconsistent results of GPER1 responses to E2. Please provide the EC50 value of E2 for GPER1 activation in the present HEK293 cells with a dose-response curve. Typically, EC50 values of GPCRs with high affinities for agonists with 200-300 Da are a few hundreds of nM. The EC50 of 1 nM would be for ERα and Erβ or their mutated ones.

2)    As requested in the previous 12th point, please correct Figure 6B by showing the circle of Pull-down-MS.

3)    Regarding the previous 14th point, I could not believe the same scales between the photos in the upper two rows of Figure 9. Such a difference in the size of cells would be questionable between healthy or infected conditions.

Author Response

please see separate file
